DATA RELEASE

# Genome assembly of the hybrid grapevine *Vitis* 'Chambourcin'

Sagar Patel[1,2,3,*], Zachary N. Harris[1,2], Jason P. Londo[4], Allison Miller[1,2,*] and Anne Fennell[5]

1 Saint Louis University, Department of Biology, 3507 Laclede Ave, St. Louis, MO 63103, USA
2 Donald Danforth Plant Science Center, 975 North Warson Road, St. Louis, MO 63132, USA
3 Eastern Virginia Medical School, School of Health Professions, Norfolk, VA 23501, USA
4 School of Integrative Plant Science, Cornell University, 630 W. North Street, Geneva, NY 14456, USA
5 South Dakota State University, Agronomy, Horticulture and Plant Science Department and BioSNTR, Brookings, SD 57006, USA

## ABSTRACT

'Chambourcin' is a French-American interspecific hybrid grape grown in the eastern and midwestern United States and used for making wine. Few genomic resources are available for hybrid grapevines like 'Chambourcin'. Here, we assembled the genome of 'Chambourcin' using PacBio HiFi long-read, Bionano optical map, and Illumina short-read sequencing technologies. We generated an assembly for 'Chambourcin' with 26 scaffolds, with an N50 length of 23.3 Mb and an estimated BUSCO completeness of 97.9%. We predicted 33,791 gene models and identified 16,056 common orthologs between 'Chambourcin', *V. vinifera* 'PN40024' 12X.v2, VCOST.v3, Shine Muscat and *V. riparia* Gloire. We found 1,606 plant transcription factors from 58 gene families. Finally, we identified 304,571 simple sequence repeats (up to six base pairs long). Our work provides the genome assembly, annotation and the protein and coding sequences of 'Chambourcin'. Our genome assembly is a valuable resource for genome comparisons, functional genomic analyses and genome-assisted breeding research.

**Subjects** Genetics and Genomics, Bioinformatics, Plant Genetics

**Submitted:** 18 January 2023

* Corresponding authors. E-mail: sgr308@gmail.com; allison.j.miller@slu.edu

Preprint submitted at https://doi.org/10.1101/2023.01.18.524616

## BACKGROUND AND CONTEXT

Grapevines (*Vitis* species) represent the world's most economically important berry-producing plants. Their fruits are used to make wine and other beverages, and are consumed as fresh or dried fruit. The European grapevine *Vitis vinifera* L. species *vinifera* is believed to have been domesticated approximately 8,000 years ago from wild populations of *V. vinifera* subspecies *sylvestris* growing in western Asia and Eastern Europe [1–3]. Grapevine growing (viticulture) spread rapidly through Europe and the Middle East, and eventually was introduced into North America as early as the mid 1700's and likely earlier [4]. In addition to the introduced *V. vinifera*, North America is home to at least 20 native *Vitis* species. Although European settlers in North America cultivated native North American *Vitis* species, few native North American grapevine species are used to make wine (e.g., *Vitis labrusca*). Despite this, many native North American *Vitis* species have become critical resources for viticulture through their use in breeding programs aimed at developing disease-resistant rootstocks and hybrid scions derived from interspecific hybridizations between wild North American *Vitis* species and cultivated European *V. vinifera*. Hybrid derivatives of crosses between North American and European grapevine

species make up a large portion of the grapevines grown in eastern and midwestern North America, and hybrid rootstocks are used throughout most grape growing regions in the world.

*Vitis* 'Chambourcin' ('Chambourcin' from here forward) is a cultivated hybrid wine grape variety derived from crosses between North American and European *Vitis* species (NCBI:txid241073). 'Chambourcin' was developed by the private breeder Joannes Seyve in France. In 1985, it was introduced into the Geneva, New York, USA repository of the United States Department of Agriculture (USDA) Agricultural Research Service. A complex hybrid, 'Chambourcin' is the product of a cross between Joannes Seyve 11369 and 'Plantet N', which includes several North American *Vitis* species in its background: *V. berlandieri* Planch., *V. labrusca* L., *V. lincecumii* Buckley, *V. riparia* Michx., *V. rupestris* Scheele, and *V. vinifera*. The full pedigree of 'Chambourcin' is publicly available [5]. 'Chambourcin' produces black-skinned berries. The flavors of wines derived from 'Chambourcin' are described as black cherry, red fruit with herbaceous notes, black pepper, and chocolate [6]. 'Chambourcin' is grown in parts of France and Australia, as well as in the United States in Colorado, Missouri, Nebraska, New Jersey, New York, Pennsylvania, and Virginia, among others.

'Chambourcin' is increasing in importance as a cultivated hybrid wine grape in the central and eastern United States. It has been used in experimental rootstock vineyards to understand rootstock effects on shoot system phenotypes [7–11], and is the parent of the new disease-resistant cultivar 'Regent'. The goals of this study were (1) to develop a high-quality reference genome for 'Chambourcin', and (2) to identify and annotate gene models for a more accurate functional genomic analysis of this disease-resistant cultivar. The work presented here advances our understanding of hybrid grapevine genomics and will facilitate the analyses of rootstock-scion interactions in 'Chambourcin' experimental vineyards.

## METHODS
### PacBio HiFi, Bionano optical map, and Illumina sequencing
'Chambourcin' leaf material was obtained from a 12-year-old experimental vineyard located at the University of Missouri Southwest Research Station in Mount Vernon, Missouri, USA. For PacBio HiFi sequencing, high molecular weight (HMW) DNA was isolated using the Nucleobond Kit (Macherey-Nagel, Bethlehem, PA, USA) following the manufacturer's protocol. Approximately 20 cg DNA was sheared to a center of mass of 10–20 kilobase (kb) in a Megaruptor 3 system. Next, a HiFi sequencing library was constructed following HiFi SMRTbell protocols for the Express Template Prep Kit 2.0 according to manufacturers' recommendations (Pacific Biosciences, California). The library was sequenced using Sequel binding and sequencing chemistry v2.0 in circular consensus sequencing (CCS) mode in a Sequel II system with a movie collection (file format of HiFi data) time of 30 h. The HiFi reads were generated with the CCS mode of pbtools [12] using a minimum Predicted Accuracy of 0.990.

For the Bionano data, DNA was isolated from fresh young leaf tissue from a 12-year-old experimental vineyard located at the University of Missouri Southwest Research Station in Mount Vernon, Missouri, USA, using the Prep™ Plant DNA Isolation kit and labeled using the Bionano Prep™ DNA Labeling Kit Direct Label and Stain (DLS) (Bionano Genomics, San Diego, California). In total, 500 ng of ultra-high molecular weight (UHMW) DNA was used



for the DLS reaction. DNA was incubated in the presence of DLE-1 Enzyme, DL-Green, and DLE-1 Buffer for 3:20 h at 37 °C. This was followed by proteinase K digestion at 50 °C for 30 min and double cleanup of the unincorporated DL-Green label. The resulting DLS sample was combined with the Flow Buffer, dithiothreitol (DTT) and DNA stain, mixed at slow speed in a rotator mixer for an hour, and then incubated overnight at 4 °C. The labeled sample was then loaded onto a Bionano flow cell in a Saphyr System for separation, imaging, and creation of digital molecules according to the manufacturer's recommendations [13]. The raw molecule set was filtered to a molecule length of 250 kb and a minimum of nine CTTAAG labels per molecule. Bionano maps were assembled without pre-assembly using the non-haplotype parameters with no Complex Multi-Path Regions (CMPR) cut and without extend-split. Bionano software (Solve, Tools and Access, v1.5.1) [14] was used for data visualization, processing, and assembly of Bionano maps. The PacBio HiFi and Bionano sequencing were done at Corteva Agriscience, Johnston, Iowa, USA.

For the Illumina whole genome data, DNA was extracted from 'Chambourcin' leaf tissue collected from the USDA Grape Germplasm Collection located in Geneva, New York. DNA was extracted using Qiagen DNeasy Plant Mini Kits (Qiagen, Valencia, California, USA) and assessed for purity and concentration using a NanoDrop spectrophotometer and Qubit fluorometer. DNA was cleaned using the Qiagen Dneasy PowerClean Pro Cleanup Kit. DNA libraries were prepared and shotgun Illumina sequenced at Novogene (San Diego, California, USA) with paired-end 150 nt reads with 40X coverage. The raw Illumina reads were trimmed with Trimmomatic (v0.39; RRID:SCR_011848) [15] using HEADCROP:4 MINLEN:70 parameters.

## Genome size estimation

The PacBio HiFi reads and 19 nt k-mers were used to estimate the genome heterozygosity using jellyfish (v2.3.0; RRID:SCR_005491) [16]. The resulting ".histo" file was visualized with GenomeScope (RRID:SCR_017014) [17].

## Genome assembly

The PacBio HiFi assembly was generated using the Hifiasm assembler (RRID:SCR_021069; v0.13-r308) [18] with default parameters. To reduce the number of small, low-coverage artifactual contigs often generated by Hifiasm [18], the assembly was filtered to exclude less than 70,000 bp contigs. The resulting HiFi contigs were merged to the DLS Bionano maps with Bionano Solve (v3.5.1) [14] using the hybridscaffold.pl script of Bionano Solve (v3.5.1) [14] to get a hybrid assembly. Each scaffold of the hybrid assembly was then checked, and small overlapping contigs were curated and removed to make a contiguous sequence. This curated diploid assembly was examined to identify alternative contigs using Purge_Haplotigs (v1.1.1; RRID:SCR_017616) [19], and the primary assembly and haplotig assemblies were created. We mapped trimmed Illumina whole genome sequences to both assemblies separately with bowtie2 (v2.3.4; RRID:SCR_016368) [20] and samtools (v1.9; RRID:SCR_002105) [21]. The resulting .bam files were used for polishing both assemblies using Pilon (v1.23; RRID:SCR_014731) [22] with one round, and the final assembly (primary assembly) and haplotig assemblies were prepared. In this study, we used only the primary assembly for all downstream analysis, but the haplotigs are maintained to cover the total heterozygous genome. Scaffolds were aligned to the *V. vinifera* 'PN40024' 12X.v2 [23]

reference genome using minimap2 (v2.17; RRID:SCR_018550) [24] and renamed based on the longest alignment with the reference genome *V. vinifera* 'PN40024' 12X.v2 chromosomes. We mapped two thousand 'Chambourcin' rhAmpSeq marker sequences [25] to the 'Chambourcin' genome assembly using the BWA aligner (v0.7.17; RRID:SCR_010910) [26]. The rhAmpSeq markers were designed to target the core *Vitis* genome and were developed from gene-rich collinear regions of 10 *Vitis* genomes [25]. These markers aid in mapping contigs on chromosomes and checking their orientation.

### Genome assembly assessment and dot plot

All assemblies generated by PacBio HiFi and Bionano data were assessed by Benchmarking Universal Single-Copy Orthologs (BUSCO) (v5.4.2; RRID:SCR_015008) [27] with genome mode and the embryophyta_odb10 dataset. The alignment of the two genomes was obtained using minimap2 (v2.17) [24] with default parameters, where the 'Chambourcin' primary assembly was considered as query while *V. vinifera* 'PN40024' 12X.v2, Shine Muscat [28], and *V. riparia* Gloire [29] were considered as the reference genome. A dot plot was obtained using the R (RRID:SCR_001905) script pafCoordsDotPlotly.R [30].

### Genome assembly analysis using k-mer spectra

The trimmed Illumina whole genome sequences were mapped separately to diploid, primary, and haplotig 'Chambourcin' assemblies using KAT (v2.4.2; RRID:SCR_016741) [31]; specifically, we used kat comp and kat plot commands.

### *De novo* gene prediction, functional annotation, and orthologous genes

*De novo* repeats were identified with RepeatModeler2 (v2.0.2a) [32], and repeats were masked by RepeatMasker (v4.1.1; RRID:SCR_012954) [33]. 'Chambourcin' RNA-seq data were downloaded from a previously published study [7] and trimmed using Trimmomatic (v0.39) [15] with HEADCROP:15 LEADING:30 TRAILING:30 MINLEN:20 parameters. The trimmed 'Chambourcin' RNA-seq reads were then mapped to the masked 'Chambourcin' primary genome assembly using HISAT2 (v2.1.0; RRID:SCR_015530) [34] and samtools (v1.9) [21] with default parameters. The resulting alignments (.bam files) and protein sequences of the *V. vinifera* 'PN40024' 12X.v2, VCost.v3 were used for gene prediction using BRAKER2 (v2.1.6; RRID:SCR_018964) [35] with –prg=gth –gth2traingenes –gff3 parameters. The resulting gene predictions (proteins, coding sequences, and annotations) were completed separately for the 'Chambourcin' primary assembly and the 'Chambourcin' haplotig assembly. The quality of the predicted proteins was assessed using BUSCO (v5.4.2) [27] with protein mode and the embryophyta_odb10 dataset. The predicted proteins of the *Vitis* 'Chambourcin' primary assembly were then functionally annotated using eggNOG-mapper (v2) (RRID:SCR_021165) [36] and related to Gene Ontology (GO), KEGG pathway, and other functional information. The GO plot was developed using the WEGO tool [37]. For the analysis of orthologous gene models, the sequences of 'Chambourcin' primary gene models, *V. vinifera* PN40024 12X.v2, VCost.v3, Shine Muscat, and *V. riparia* Gloire were analyzed using OrthoVenn2 [38] with default settings, *E*-value: $1 \times 10^{-5}$, and inflation value: 1.5.



### Plant transcription factors prediction, phylogenetic tree, and WRKY classification

The plant transcription factors for the gene models of the 'Chambourcin' primary assembly, *V. vinifera* PN40024 12X.v2, and VCost.v3 were identified using the Plant Transcription Factor Database (PlantTFDB v5.0; RRID:SCR_003362) [39]. The identified transcription factors were divided into subfamilies according to their sequence relationship with *V. vinifera*. For the circular phylogenetic tree and WRKY sequences of 'Chambourcin' primary gene models and *V. vinifera* PN40024 12X.v2, VCost.v3 gene models retrieved from PlantTFDB (5.0) [39] and aligned using ClustalW method in MEGA7 [40]. A phylogenetic analysis was carried out using the neighbor-joining method with 1,000 bootstrap replications, and the evolutionary distances were computed using the Poisson correction method with the Pairwise Deletion option. The WRKY classification of 'Chambourcin' primary gene models was carried out using the same method described in a previous study [41].

### Synteny, Simple Sequence Repeats (SSRs), and Circos plot

The 'Chambourcin' masked primary genome assembly and gene annotations were aligned to *V. vinifera* 'PN40024' 12X.v2, Shine Muscat, and *V. riparia* 'Gloire' genomes and gene annotations separately using the 'promer' option of the MUMmer program in SyMAP (v4.2) [42]. We used MIcroSAtellite [43] to find SSRs in the unmasked 'Chambourcin' primary genome assembly. The Circos plots were generated using circos (v0.69.6; RRID:SCR_011798) [44] with the 'Chambourcin' primary genome assembly, SSRs, and the 'Chambourcin' primary gene annotations.

### Mapping of Illumina whole genome reads, and RNA-seq reads to the genome assembly

The trimmed Illumina whole genome sequences were mapped to the diploid, primary, and haplotig 'Chambourcin' assemblies separately with bowtie2 (v2.3.4) [20] and samtools (v1.9) [21]. The resulting .bam files were used to obtain mapping results using the samtools flagstat [21] command. We also mapped the trimmed 'Chambourcin' RNA-seq reads [7] to diploid, primary, and haplotig 'Chambourcin' assemblies separately using HISAT2 (v2.1.0) [34] and samtools (v1.9) [21]. The resulting .bam files were used to obtain mapping results using the samtools flagstat [21] command.

### RESULTS AND DISCUSSION

### Genome sequencing and assembly of 'Chambourcin'

We generated a high-quality and contiguous genome sequence of 'Chambourcin' using PacBio HiFi Sequencing, Bionano third-generation DNA sequencing, and Illumina short-read sequencing. A total of 1,634,814 PacBio HiFi filtered reads was produced with an average length of 16,148 bp and genome coverage of 28X. The filtered Bionano data resulted in a subset of 1,243,428 molecules with a total length of 429,808.857 Mbp and coverage of 188.70X. In total, 124 Bionano maps, with a total length of 962.964 Mbp and an N50 of 13,725 bp, were assembled, corresponding to the diploid complement. A total of 154,152,068 filtered Illumina short reads and genome coverage of 40X were generated for genome polishing. We estimated heterozygosity to be 2.28% in the 'Chambourcin' genome (Figure 1), which is higher than estimates for heterozygosity in any of the other *Vitis* genomes

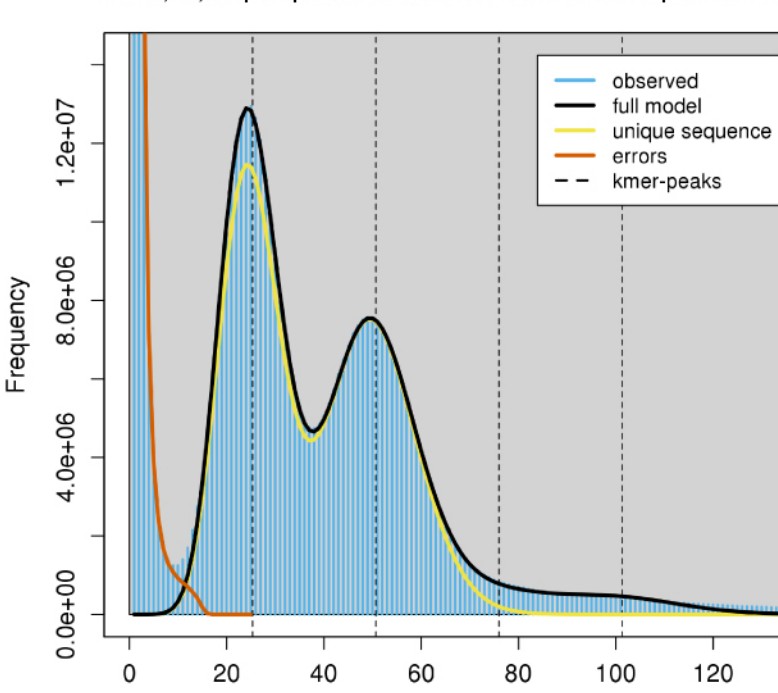

**Figure 1.** GenomeScope plot estimating the heterozygosity of *Vitis* 'Chambourcin'.

sequenced to date [23, 28, 29]. Relatively higher levels of heterozygosity in the 'Chambourcin' genome compared to other *Vitis* species are expected, given the complex interspecific pedigree of this cultivar. A GenomeScope plot of clean reads demonstrated two peaks of coverage; the first peak located at 25X coverage corresponds to the heterozygous portion of the genome, and the second peak at 52X coverage corresponds to the homozygous portion of the genome (Figure 1).

A *de novo* 'Chambourcin' genome was assembled using HiFi, Bionano, and Illumina data. First, a contig assembly of the PacBio HiFi reads resolved the reads into 196 contigs with an N50 of 12,215,205 bp and a total length of 949,347,381 bp (Table 1). The PacBio HiFi contig assembly was then merged with the Bionano maps to get an initial hybrid assembly comprising 67 scaffolds with an N50 length of 16,400,326 bp, a maximum scaffold length of 39,458,994 bp, and a total scaffold length of 903,810,753 bp (Table 1). After manual curation, the hybrid assembly included 64 scaffolds with an N50 length of 16,278,793 bp, a maximum length of 39,458,994 bp, and a total length of 869,222,201 bp (Table 1). The hybrid assembly was partitioned into a final primary assembly (493,554,689 bp) and a haplotig assembly (375,458,233 bp) (Table 1). Pilon (v1.23) corrected 19,771 single nucleotide polymorphisms (SNPs), 636 ambiguous bases, 16,469 small insertions totaling 121,315 bases, and 21,889 small deletions totaling 129,590 bases in the primary assembly; Pilon (v1.23) also corrected 21,394 SNPs, 606 ambiguous bases, 14,999 small insertions totaling 102,126 bases, and 18,929 small deletions totaling 105,957 bases in the haplotig assembly. After polishing for 'Chambourcin', the final primary assembly contained 26 scaffolds with an N50 length of

**Table 1.** Descriptive statistics and BUSCO results for the 'Chambourcin' genome assembly.

| Details | PacBio HiFi assembly | Hybrid assembly (PacBio HiFi + Bionano) | Hybrid assembly (curated) | Final primary assembly | Final haplotig assembly |
|---|---|---|---|---|---|
| | | | | (after purge haplotigs) | |
| Genome assembly results | | | | | |
| Number of scaffolds | 196 | 67 | 64 | 26 | 38 |
| Total size of scaffolds | 949,347,381 | 903,810,753 | 869,222,201 | 493,554,689 | 375,458,233 |
| Longest scaffold | 47,120,234 | 39,458,994 | 39,458,994 | 39,456,434 | 28,439,729 |
| Shortest scaffold | 70,924 | 1,571,397 | 1,571,397 | 5,519,669 | 1,571,459 |
| Number of scaffolds > 1M nt | 131 | 67 | 64 | 26 | 38 |
| Number of scaffolds > 10M nt | 27 | 40 | 38 | 22 | 16 |
| N50 scaffold length | 12,215,205 | 16,400,326 | 16,278,793 | 23,325,629 | 12,462,019 |
| scaffold %N | 0 | 0.83 | 0.76 | 0.02 | 1.71 |
| Number of contigs | 196 | 1,365 | 1,194 | 44 | 136 |
| Total size of contigs | 949,347,381 | 896,337,746 | 862,647,977 | 493,450,270 | 369,041,914 |
| Longest contigs | 47,120,234 | 32,988,011 | 32,988,011 | 32,983,396 | 22,801,634 |
| Shortest contigs | 70,924 | 6 | 6 | 6 | 6 |
| Number of contigs > 1M nt | 131 | 100 | 95 | 36 | 59 |
| Number of contigs > 10M nt | 27 | 33 | 31 | 22 | 9 |
| N50 contigs length | 12,215,205 | 14,022,606 | 13,473,621 | 16,713,841 | 7,864,215 |
| contig %N | 0 | 0 | 0 | 0 | 0 |
| BUSCO results | | | | | |
| Complete BUSCOs (C) = (S) + (D) | 1,593 (98.7%) | 1,594 (98.7%) | 1,592 (98.6%) | 1,580 (97.9%) | 1,180 (73.1%) |
| Complete and single-copy BUSCOs (S) | 121 (7.5%) | 367 (22.7%) | 480 (29.7%) | 1,546 (95.8%) | 1,140 (70.6%) |
| Complete and duplicated BUSCOs (D) | 1,472 (91.2%) | 1,227 (76%) | 1,112 (68.9%) | 34 (2.1%) | 40 (2.5%) |
| Fragmented BUSCOs (F) | 13 (0.8%) | 13 (0.8%) | 14 (0.9%) | 17 (1.1%) | 23 (1.4%) |
| Missing BUSCOs (M) | 8 (0.5%) | 7 (0.5%) | 8 (0.5%) | 17 (1%) | 411 (25.5%) |
| Total BUSCOs | 1,614 | 1,614 | 1,614 | 1,614 | 1,614 |

23,325,629 bp, and the longest scaffold measured 39,456,434 bp (Table 1). The secondary haplotig assembly after polishing contained 38 haplotig scaffolds with an N50 length of 1,2462,019 bp, and the longest scaffold measured 28,439,729 bp (Table 1). We identified 97.9% complete BUSCOs for primary genome assembly and 73.1% complete BUSCOs for haplotig genome assembly (Table 1).

The 'Chambourcin' primary genome assembly was aligned to the reference genomes *V. vinifera* 'PN40024' 12X.v2 [23] (see Table 1 in GigaDB [45]), Shine Muscat [28], and *V. riparia* 'Gloire' [29]. A dot plot was generated to facilitate the comparisons among genomes. Collinearity between 'Chambourcin' and *V. vinifera* 'PN40024' 12X.v2, Shine Muscat, and *V. riparia* 'Gloire' was observed as a straight diagonal line without large gaps in the dot plot, confirming the high synteny of the 'Chambourcin' genome with *V. vinifera* 'PN40024' 12X.v2 (Figure 2A), Shine Muscat (Figure 2B), and *V. riparia* 'Gloire' (Figure 2C). To further validate the 'Chambourcin' genome assembly, we mapped 'Chambourcin' rhAmpSeq markers [25] to the 'Chambourcin' genome assembly. We found 99% of rhAmpSeq markers mapped to 'Chambourcin' scaffolds and mapped to the same chromosomes and positions the markers were derived from in the collinear *Vitis* core genome (see Table 2 in GigaDB [45]).

Synteny analyses of the 'Chambourcin' primary genome assembly with *V. vinifera* 'PN40024' 12X.v2, Shine Muscat, and *V. riparia* 'Gloire' genomes were used to identify syntenic blocks between species. The 'Chambourcin' primary assembly scaffolds aligned with larger syntenic blocks and covered the whole chromosomes of *V. vinifera* PN40024 12X.v2 (Figure 2D), Shine Muscat (Figure 2D), and *V. riparia* 'Gloire' (Figure 2E). This alignment of the primary genome with *V. vinifera* PN40024 12X.v2, Shine Muscat, and

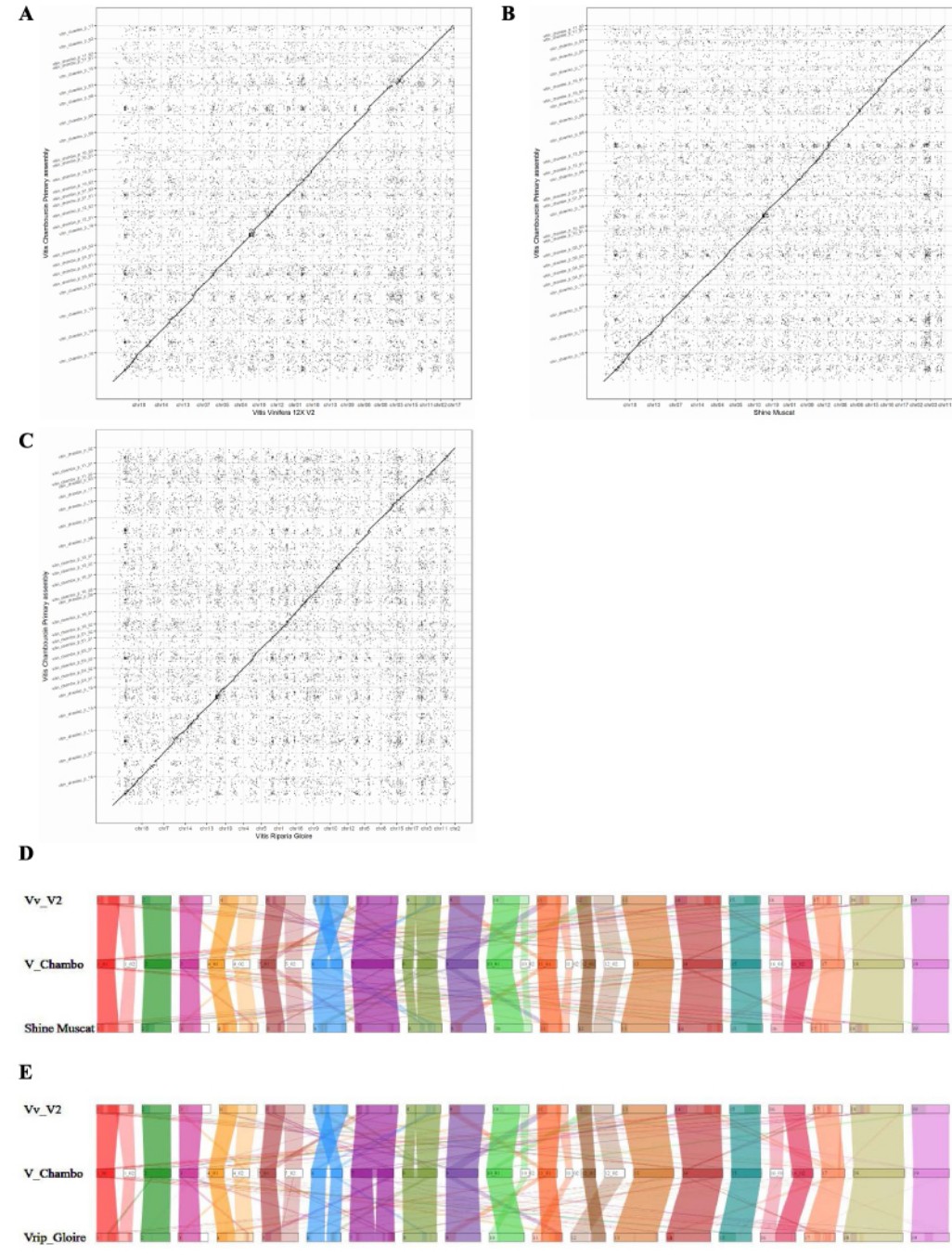

**Figure 2.** Comparative study of the 'Chambourcin' genome assembly. (A) Dotplot of the 'Chambourcin' primary genome assembly and *V. vinifera* 'PN40024' 12X.v2. (B) Dotplot of the 'Chambourcin' primary genome assembly and Shine Muscat. (C) Dotplot of the 'Chambourcin' primary genome assembly and *V. riparia* Gloire. (D) Synteny between 'Chambourcin' primary genome assembly, *V. vinifera* PN40024 12X.v2 genome, and Shine Muscat. (E) Synteny between 'Chambourcin' primary genome assembly, *V. vinifera* PN40024 12X.v2 genome, and *V. riparia* Gloire genome.

*V. riparia* 'Gloire' indicated highly contiguous 'Chambourcin' scaffolds useful for comparative genomic analyses.

## Genome assembly analysis using K-mer spectra plot

The genome quality was assessed with Illumina whole genome reads separately for diploid, primary, and haplotig genome assemblies using KAT tool [31] and K-mer spectra plots were generated. A K-mer spectra is a graphical representation showing how many k-mers appear a certain number of times. The frequency of occurrence is plotted on the *x*-axis and the number of k-mers is plotted on the *y*-axis. All K-mer spectra plots for 'Chambourcin' diploid, primary and haplotig assemblies were identified with an error distribution under 10X, a heterozygous peak at 35X, and a homozygous peak at 67X (Figure 3A–C). The different colors in the K-mer spectra plot shows the different occurrences of k-mers. The black color represents read content occurs at zero time (0X), the red color represents unique content occurs at one time (1X), the purple color represents content occurs at two times (2X) and the green color represents content occurs at three times (3X). The K-mer spectra plot of the diploid genome assembly shows that the read content shown in black color is absent from the assembly, and red peak occurs once, showing most of the heterozygous content. At the same time, purple peak indicates more duplications on homozygous content (Figure 3A). The K-mer spectra plot of the primary genome is more collapsed, including mostly a single copy of the homozygous content and less of the heterozygous content (Figure 3B). The K-mer spectra plot for the haplotig genome assembly identified two black peaks representing read content in both the heterozygous and homozygous regions (Figure 3C). These K-mer spectra plots provides useful information for genome assembly assessment using whole genome short reads to identify duplicate regions in the assembly and visualize the genome assembly. This visualization is useful for genome assembly curation steps to identify accurate primary and haplotig assembly from a diploid genome assembly.

## Repeat sequence annotation

Repeated regions were binned into seven different classes: long interspersed nuclear elements (LINEs) (4.43%), long terminal repeats (LTRs) (15.66%), DNA transposons (2.03%), rolling-circles (0.58%), low complexity repeats (0.37%), simple repeats (1.21%), and unclassified repeats (31.95%) (Figure 4; Table 2). The repetitive sequence content in the 'Chambourcin' primary genome assembly (56.23%) was higher than previously reported for *V. riparia* 'Manitoba 37' (46%) [41], *V. vinifera* 'PN40024' 12X.v2 (35.12%) [23], Shine Muscat (48%) [28], and *V. riparia* 'Gloire' (33.94%) [29]. SSRs are tandem repeats of DNA that have been used to develop robust genetic markers. We identified 304,571 SSRs, repeating units of 1–6 base pairs in length, in the 'Chambourcin' primary genome assembly (Figure 4; and see GigaDB Table 3 [45]).

## Gene annotation and orthologous genes

A total of 33,791 gene models were predicted for the 'Chambourcin' primary genome assembly (Figure 4). We identified 94.6% complete BUSCOs (C); of these, 86.9% were designated single-copy BUSCOs (S), and 7.7% were designated duplicated BUSCOs (D) (Table 3). As evidenced by the high number of complete single-copy genes identified, the BUSCO results indicate that the 'Chambourcin' primary genome assembly offers

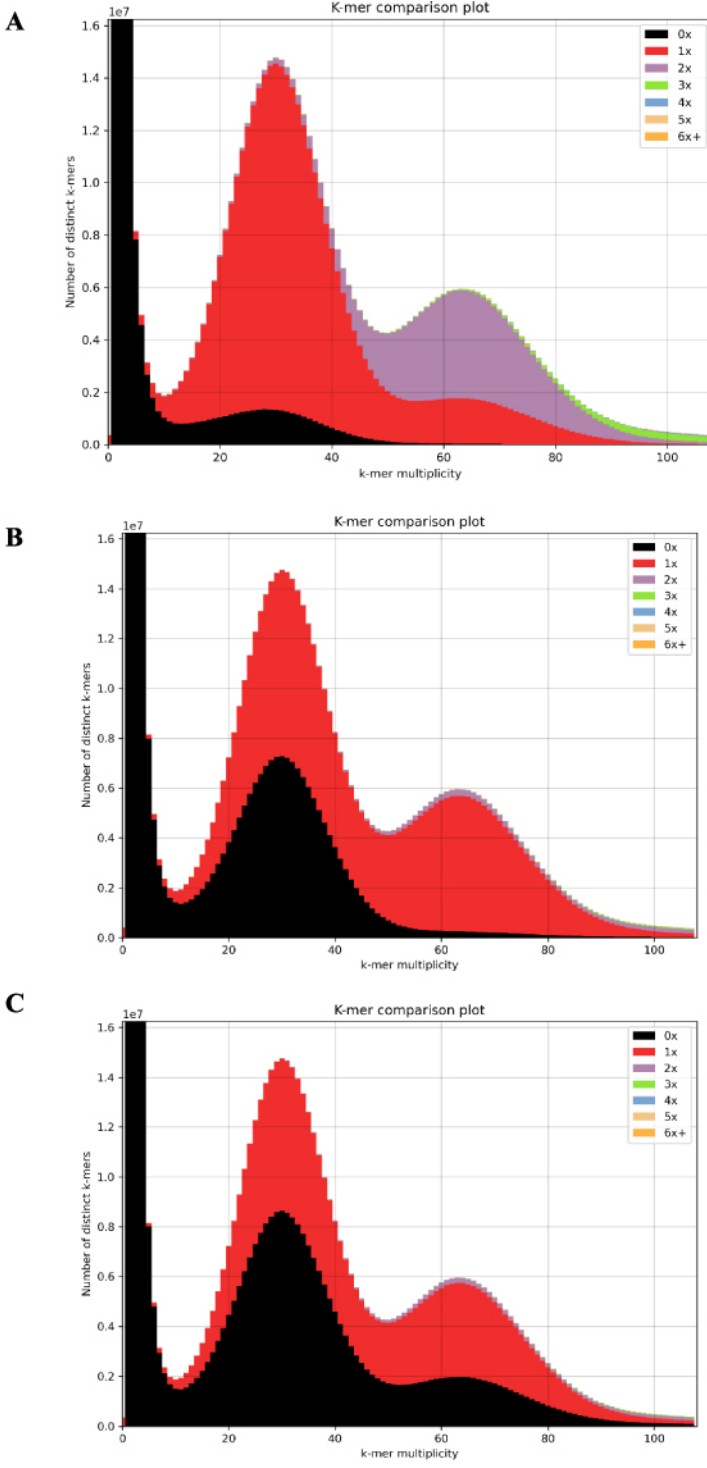

**Figure 3.** KAT k-mer spectra plot. (A) k-mer spectra plot for the diploid genome assembly. (B) k-mer spectra plot for the primary genome assembly. (C) k-mer spectra plot for the haplotig genome assembly.

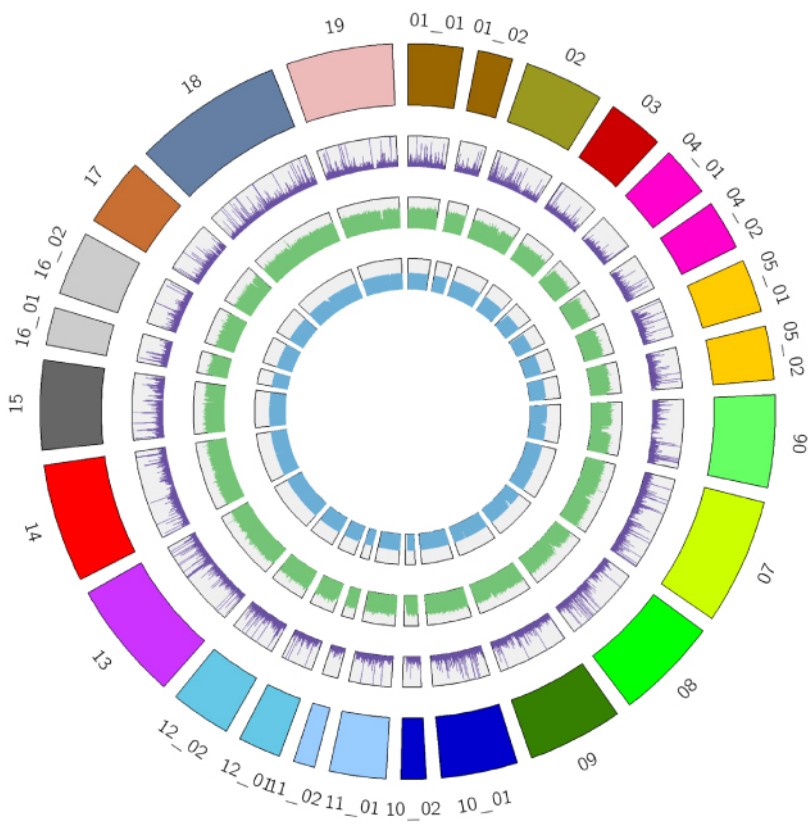

**Figure 4.** Circos plot. The outer ring represents all scaffolds of 'Chambourcin' primary genome assembly in different colors. The second ring (purple) represents the SSRs. The third ring (green) represents the repetitive sequences. The fourth ring (blue) represents the gene annotations.

**Table 2.** Repetitive sequences in the 'Chambourcin' genome assembly.

| Details | Primary assembly | Haplotig assembly |
|---|---|---|
| Total length | 493,554,689 bp | 375,458,233 bp |
| Bases masked | 277,503,542 bp (56.23%) | 211,395,579 bp (56.30%) |
| Retroelements: | 99,122,846 bp (20.08%) | 82,725,835 bp (22.03%) |
| LINEs: | 21,852,466 bp (4.43%) | 15,553,322 bp (4.14%) |
| RTE/Bov-B | 180,809 bp (0.04%) | 133,053 bp (0.04%) |
| L1/CIN4 | 21,671,657 bp (4.39%) | 15,420,269 bp (4.11%) |
| LTR elements: | 77,270,380 bp (15.66%) | 67,172,513 bp (17.89%) |
| Ty1/Copia | 39,408,750 bp (7.98%) | 32,756,463 bp (8.72%) |
| Gypsy/DIRS1 | 33,756,474 bp (6.84%) | 31,170,356 bp (8.30%) |
| DNA transposons: | 9,995,420 bp (2.03%) | 7,711,872 bp (2.05%) |
| hobo-Activator | 3,826,157 bp (0.78%) | 2,619,712 bp (0.70%) |
| Tourist/Harbinger | 640,576 bp (0.13%) | 454,161 bp (0.12%) |
| Rolling-circles | 2,864,922 bp (0.58%) | 2,386,306 bp (0.64%) |
| Unclassified | 157,704,064 bp (31.95%) | 112,960,798 bp (30.09%) |
| Total interspersed repeats | 266,822,330 bp (54.06%) | 203,398,505 bp (54.17%) |
| Simple repeats | 5,980,823 bp (1.21%) | 4,443,253 bp (1.18%) |
| Low complexity | 1,835,467 bp (0.37%) | 1,167,515 bp (0.31%) |

comprehensive coverage of the expected gene space. Functional annotation of the 'Chambourcin' primary gene models (33,791) was done using the EggNOG database

**Table 3.** 'Chambourcin' gene prediction (Coding Sequences (CDS) and protein sequences) and BUSCO results of protein sequences.

| Details | Primary assembly | Haplotig assembly |
|---|---|---|
| Gene prediction results | | |
| Total CDS and protein | 33,791 | 24,018 |
| Total CDS (bp) | 36,761,139 | 25,814,082 |
| Mean CDS (bp) | 1,087.9 | 1,074.8 |
| Longest CDS (bp) | 15,867 | 21,819 |
| Total protein (bp) | 12,219,929 | 8,580,681 |
| Mean protein (bp) | 361.6 | 357.3 |
| Longest protein (bp) | 5,288 | 7,272 |
| BUSCO results | | |
| Complete BUSCOs (C) = (S) + (D) | 1,528 (94.6%) | 1,136 (70.3%) |
| Complete and single-copy BUSCOs (S) | 1,403 (86.9%) | 1,032 (63.9%) |
| Complete and duplicated BUSCOs (D) | 125 (7.7%) | 104 (6.4%) |
| Fragmented BUSCOs (F) | 50 (3.1%) | 31 (1.9%) |
| Missing BUSCOs (M) | 36 (2.3%) | 447 (27.8%) |
| Total BUSCOs | 1,614 | 1,614 |

(see GigaDB Table 4 [45]). A total of 27,075 'Chambourcin' primary proteins were annotated, and 86% (22,977) of these proteins were annotated with *V. vinifera* V1 gene models (see GigaDB Table 4 [45]). Out of the 27,075 'Chambourcin' annotated primary proteins, 13,311 gene models were identified with GO accessions and further classified into three sub-ontologies: biological process (11,399), cellular component (11,472), and molecular function (9,977) (Figure 5F) (GigaDB Table 4 [45]). A total of 8,460 'Chambourcin' primary proteins were annotated with KEGG pathways (GigaDB Table 4 [45]). Using OrthoVenn2, we identified 16,056 common orthologs between 'Chambourcin' primary gene models, *V. vinifera* PN40024 12X.v2 annotation, Shine Muscat, and *V. riparia* 'Gloire' (Figure 5A). In total, 16,476 orthologous gene models were found between the 'Chambourcin', Shine Muscat, and *V. riparia* 'Gloire' (Figure 5B). Finally, 19,477 gene models were orthologous with *V. vinifera* PN40024 12X.v2 VCost.v3 proteins (Figure 5C), 18,669 gene models were orthologous with Shine Muscat (Figure 5D), and 18,183 gene models were orthologous with *V. riparia* 'Gloire' (Figure 5E).

## Plant transcription factors and 'Chambourcin' WRKY transcription factor classification

Using PlantTFDB 5.0, 1,606 plant transcription factors representing 58 gene families were identified from 'Chambourcin' primary proteins (see GigaDB Table 5 [45]). A similar number of transcription factors was identified for the AP2, NAC, RAV, and WRKY gene families, as found in *V. vinifera* 'PN40024' 12X.v2, VCost.v3. We identified 65 WRKY sequences in 'Chambourcin' and 62 in *V. vinifera* PN40024 12X.v2, VCost.v3 (Figure 6) (Table 4; GigaDB Table 5 [45]). WRKY transcription factors regulate many processes in plants and algae, such as the responses to biotic and abiotic stresses and seed dormancy. The 'Chambourcin' WRKY subfamily classification was similar to *V. vinifera* 'PN40024' 12X.v2 and *V. riparia* 'Manitoba 37' (Table 4). These results show the high coverage of 'Chambourcin' primary proteins.

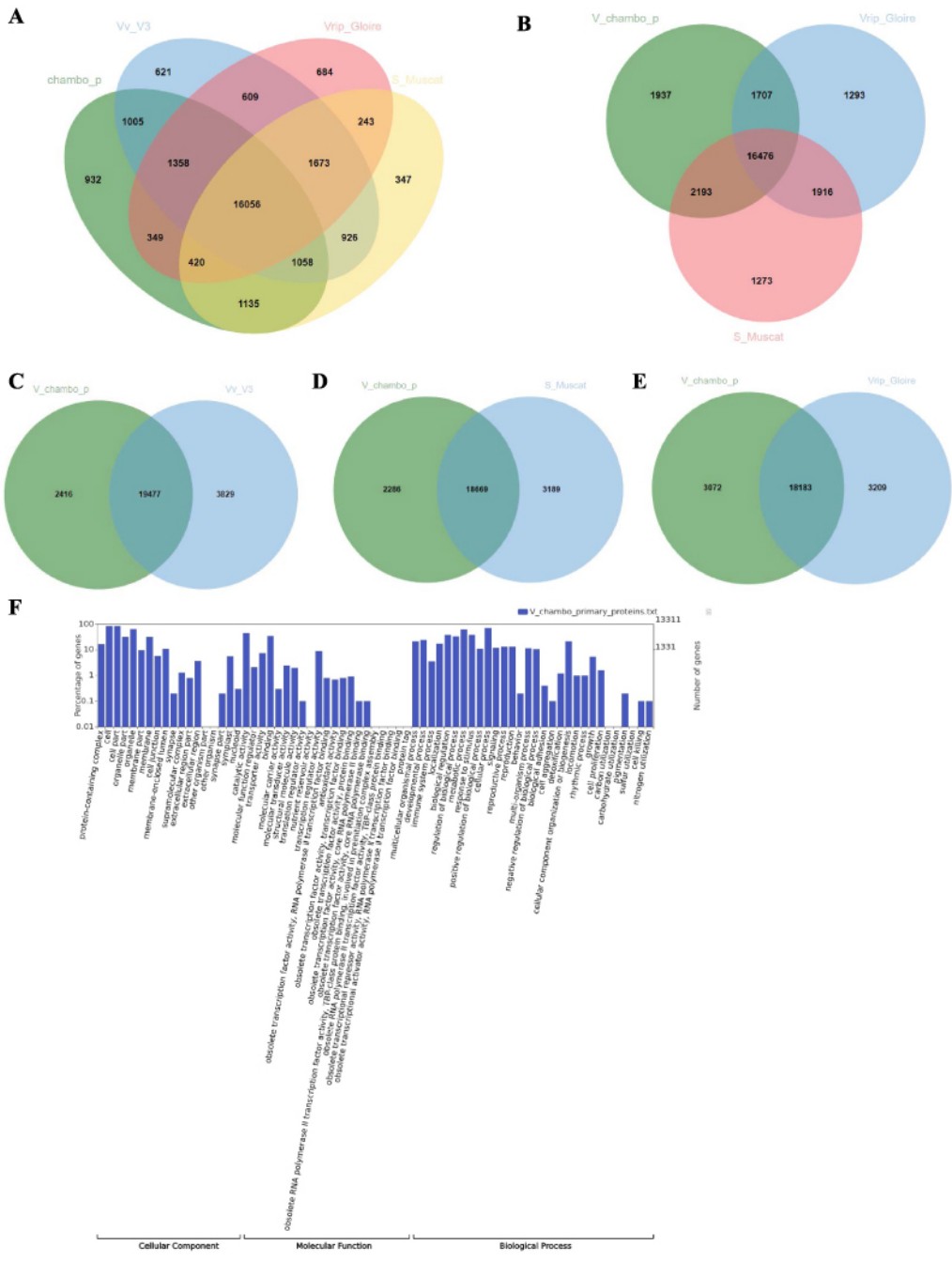

**Figure 5.** Venn diagram of 'Chambourcin' primary proteins with other grapevine species. (A) Venn diagram of orthologous genes in the 'Chambourcin' primary proteins, *V. vinifera* PN40024 12X.v2, VCost.v3, Shine Muscat, and *V. riparia* Gloire. (B) Orthologous genes in the 'Chambourcin' primary proteins, Shine Muscat, and *V. riparia* Gloire. (C) Orthologous genes in 'Chambourcin' primary proteins and *V. vinifera* PN40024 12X.V3 proteins. (D) Orthologous genes in 'Chambourcin' primary proteins and Shine Muscat. (E) Orthologous genes in 'Chambourcin' primary proteins and *V. riparia* Gloire species. (F) GO results for 'Chambourcin' primary proteins.

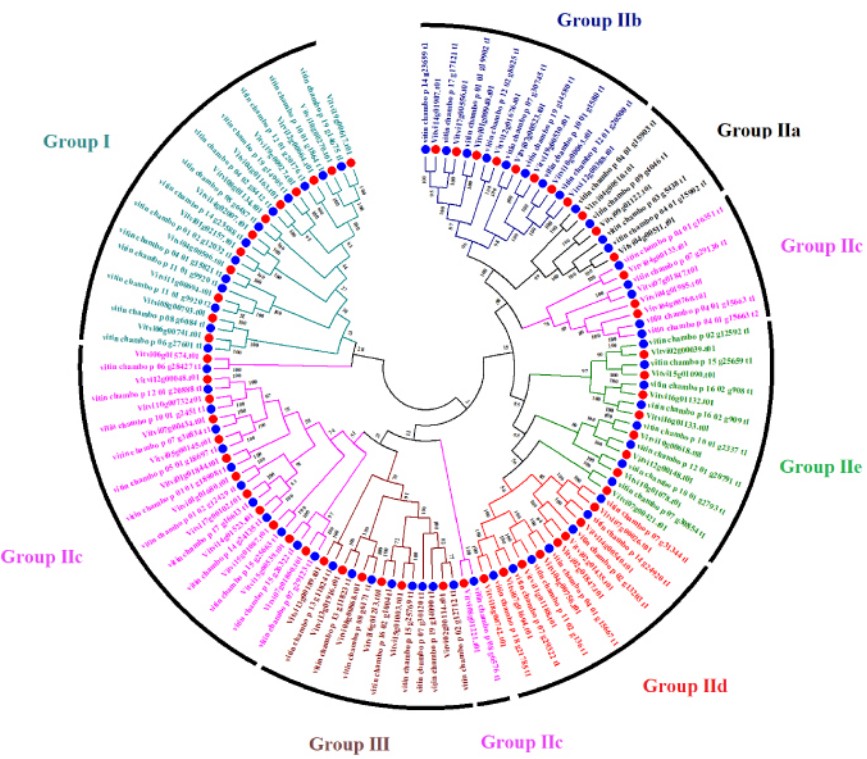

**Figure 6.** 'Chambourcin' and *V. vinifera* PN40024 12X.v2, VCost.v3 WRKY transcription factors. The blue dots represent 'Chambourcin' and the red dots represent *V. vinifera* PN40024 12X.v2, VCost.v3. The bootstrap values displayed are at the nodes.

**Table 4.** Comparison of the WRKY transcription factor classification for 'Chambourcin' with other grape species.

| Species | Group I | Group II | | | | | Group III | Total |
|---|---|---|---|---|---|---|---|---|
| | | IIa | IIb | IIc | IId | IIe | | |
| Chambourcin | 13 | 4 | 8 | 17 | 7 | 8 | 8 | 65 |
| *V. vinifera* V3 | 12 | 3 | 8 | 17 | 8 | 8 | 6 | 62 |
| *V. riparia** | 13 | 3 | 8 | 19 | 8 | 9 | 7 | 67 |

* (Patel *et al.*, 2020) [41].

## Mapping of Illumina whole genome reads, and RNA-seq reads to the genome assembly

We aligned trimmed Illumina whole genome reads to 'Chambourcin' diploid, primary, and haplotig assemblies separately and obtained an average of 95.08%, 90.17%, and 75.88% mapping results, respectively (Table 5). We also separately mapped trimmed RNA-seq reads to 'Chambourcin' diploid, primary, and haplotig assemblies and obtained 87%, 80.81%, and 60.77% mapping results, respectively. The mapping results of both trimmed Illumina whole genome and trimmed RNA-seq reads to genome assemblies show that most reads were mapped to the diploid assembly, followed by primary and haplotig assemblies. The mapping results for the primary genome assembly retained most of the genome from the diploid assembly, while the smallest number of mapped reads belonged to the haplotig assembly. These results suggest that the smallest genome portion is missing in the haplotig assembly.

**Table 5.** Mapping of Illumina whole genome reads to the genome assembly.

| Illumina reads | Diploid assembly | Primary assembly | Haplotig assembly |
|---|---|---|---|
| HF3YFDSXX<br>Total reads: 105,912,898 | 101,506,499<br>(mapped 95.84%)<br>99,659,968<br>(properly paired 94.10%) | 96,261,693<br>(mapped 90.89%)<br>92,943,104<br>(properly paired 87.75%) | 81,153,020<br>(mapped 76.62%)<br>77,352,356<br>(properly paired 73.03%) |
| HJNMYDSXX<br>Total reads: 18,428,676 | 17,632,525<br>(mapped 95.68%)<br>17,318,958<br>(properly paired 93.98%) | 16,719,389<br>(mapped 90.72%)<br>16,148,280<br>(properly paired 87.63%) | 14,095,158<br>(mapped 76.48%)<br>13,438,716<br>(properly paired 72.92%) |
| HL7HHDSXX<br>Total reads: 12,705,214 | 12,193,364<br>(mapped 95.97%)<br>11,974,672<br>(properly paired 94.25%) | 11,559,561<br>(mapped 90.98%)<br>11,164,510<br>(properly paired 87.87%) | 9,734,853<br>(mapped 76.62%)<br>9,285,522<br>(properly paired 73.08%) |
| HMMHLDSXX<br>Total reads: 171,257,348 | 159,008,113<br>(mapped 92.85%)<br>147,418,324<br>(properly paired 86.08%) | 150,914,465<br>(mapped 88.12%)<br>136,316,148<br>(properly paired 79.60%) | 126,442,365<br>(mapped 73.83%)<br>112,044,416<br>(properly paired 65.42%) |

## CONCLUSION

In this study, we presented the genome assembly of 'Chambourcin', a complex interspecific hybrid grape cultivar, using PacBio HiFi long read sequencing, Bionano third-generation sequencing data, and Illumina short read data. The comparative genomic analyses of 'Chambourcin' with the reference genome of *V. vinifera* 'PN40024' 12X.v2, Shine Muscat, and *V. riparia* 'Gloire' indicated that the 'Chambourcin' genome aligns well with other grape genomes without any large structural variation. Ortholog analyses of the 'Chambourcin' primary gene models, *V. vinifera* 'PN40024' 12X.v2, VCost.v3, Shine Muscat, and *V. riparia* 'Gloire', revealed that our 'Chambourcin' genome assembly and gene annotations are a high-quality grapevine resource for the research community.

Interspecific hybrids derived from two or more *Vitis* species are common in nature [1]. They are the cornerstone of grapevine rootstocks grown worldwide, cultivars that predominate in eastern and midwestern North America, and new disease-resistant genotypes currently in development [46]. The sequencing data, scaffold assemblies, and gene annotations of the 'Chambourcin' genome assembly described here provide a valuable resource for genome comparisons, functional genomic analyses, and genome-assisted breeding research.

## DATA AVAILABILITY

The PacBio HiFi and Illumina whole genome reads are deposited in the NCBI BioProject with accession PRJNA754438. The Sequence Read Archive (SRA) accession of the PacBio HiFi reads is SRR15530464, and the SRA accession of the Illumina whole genome reads are SRR24093946, SRR24093988, SRR24095403, and SRR24097763. The Bionano maps, genome assembly, gene annotation, proteins, and other data are available on figshare [47]. Supplementary tables and additional data is in the GigaDB repository [45].

## LIST OF ABBREVIATIONS

BUSCO: Benchmarking Universal Single-Copy Orthologs; CCS: circular consensus sequencing; DLS: Direct Label and Stain; GO: Gene Ontology; HMW: high molecular weight; LINEs: long interspersed nuclear elements; LTRs: long terminal repeats; CDS: Coding Sequences; PlantTFDB: Plant Transcription Factor Database; SNPs: single nucleotide polymorphisms; SSRs: Simple Sequence Repeats; USDA: United States Department of Agriculture.

## DECLARATIONS

### Ethics approval

The authors declare that ethical approval was not required for this type of research.

### Competing Interests

The authors declare that the research was conducted without any commercial or financial relationships that could be construed as a potential conflict of interest.

### Authors' contributions

SP, ZH, AF, and AM conceived and designed this study. AF provided computational resources and guidance, and JPL provided 'Chambourcin' samples for the Illumina whole genome sequencing and rhAmpseq marker haplotype sequences. SP processed the DNA sequences for 'Chambourcin', assembled the genome, and conducted the synteny analysis.
SP processed the RNA-Seq data for gene prediction, and conducted gene prediction and annotation. SP conducted comparative genomics analyses and uploaded the sequences to NCBI and figshare. SP wrote the first draft of the manuscript. SP, AM, ZH, AF, and JPL reviewed and finalized the manuscript.

### Funding

This project was funded by NSF Plant Genome Research Program 1546869 to AM, AF, and JPL.

### Acknowledgements

We acknowledge Laszlo Kovacs for collecting 'Chambourcin' samples for sequencing and Alex Harkess for assistance in developing protocols for DNA extractions from 'Chambourcin'. Roberto Villegas-Diaz, Chad Julius, Luke Grassman, and Rachael Auch assisted with installing and debugging tools in the South Dakota University Research Cyberinfrastructure High Performance Computing Cluster Roaring Thunder.

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
