## [Editor Report]

Comments to the AuthorHybrid genomes are tricky to assemble, and few genomic resources are available for hybrid grapevines such as ‘Chambourcin’, a French-American interspecific hybrid grape grown in the eastern and midwestern United States. Here is an attempt to assemble Chambourcin’ using a combination of PacBio HiFi long-reads, Bionano optical maps, and Illumina short-read sequencing technologies. Producing an assembly for with 26 scaffolds, an N50 length 23.3 Mb and an estimated BUSCO completeness of 97.9% that can be used for genome comparisons, functional genomic analyses, and genome-assisted breeding research. Error correction and pilon polishing was a challenge with this hybrid assembly, but after trying a few different approaches in the review process have improved it, and as they have documented what they did and are clear about the final metrics, users can assess the quality themselves.

---

## [Reviewer Report]

Comments on revised manuscriptSince the authors have made some correction and improvement, the genome quality was still low, and the manuscript has not improvement significantly. Authors should provide the haplotype sequences, and describe the genome assembly and correction steps more clearly. Moreover, the innovation of the article is insufficient. I suggest reject.

---

## [Reviewer Report]

Comments on revised manuscriptEfforts to improve the accuracy of the MS and the availability of data are clear in the revised version. Authors have included descriptions of M&M procedures and information about the origin of several datasets that were missing. They also included files with commands and original results to the FTP server. In addition, they did further de-duplication of the assembly, added Illumina sequencing for assembly polishing, and included further QC stats and comparisons to another recently published hybrid grapevine genome assembly.  Most revision actions were successful. However, it is not recommended to polish HiFi assemblies with Illumina reads as in most cases it harms the consensus quality more than it improves it, which is particularly true for repetitive and highly heterozygous genomes like the one of Chambourcin grapevine cultivar. In fact, the BUSCO Completeness of 97.9% after Pilon short-read polishing compared to 98.2% in the former version indicates that polishing with Illumina short-reads is indeed harming in this revised version. I indeed agree with authors that 28x depth of PacBio HiFi reads should suffice to produce a quality genome assembly without using more depth or another sequencing technologies as they indicate in their response. I would recommend to remove the Pilon polishing from the final assembly version, which is only recommended in error-prone PacBio CLR or Nanopore assemblies. Instead, authors could use the Illumina reads for k-mer analysis of assembly consensus quality and completeness.

---

## [Reviewer Report]

Reviewer name and names of any other individual's who aided in reviewer Lingfei ShangguanDo you understand and agree to our policy of having open and named reviews, and having your review included with the published papers. (If no, please inform the editor that you cannot review this manuscript.)YesIs the language of sufficient quality?YesPlease add additional comments on language quality to clarify if needed
Are all data available and do they match the descriptions in the paper? YesAdditional CommentsAre the data and metadata consistent with relevant minimum information or reporting standards? See GigaDB checklists for examples <a href="http://gigadb.org/site/guide" target="_blank">http://gigadb.org/site/guide</a>YesAdditional CommentsIs the data acquisition clear, complete and methodologically sound?NoAdditional CommentsIs there sufficient detail in the methods and data-processing steps to allow reproduction?YesAdditional CommentsIs there sufficient data validation and statistical analyses of data quality? YesAdditional CommentsIs the validation suitable for this type of data?YesAdditional CommentsIs there sufficient information for others to reuse this dataset or integrate it with other data?YesAdditional CommentsAny Additional Overall Comments to the AuthorGrapevine is one of the most important fruit crops in the world, and ‘Chambourcin’ is a hybrid wine grape variety in the world, which represented the cross species between North American and European Vitis species. The authors have sequenced the genome sequence of ‘Chambourcin’, and obtained the repeat sequences and gene annotation information.  However, the sequence depth was too low for the grape genome, especially the high heterozygosity. They also not applied the illumine sequencing for sequence correction. RecommendationReject (Unsound or Unusuable)

---

## [Reviewer Report]

Reviewer name and names of any other individual's who aided in reviewer Pablo Carbonell-BejeranoDo you understand and agree to our policy of having open and named reviews, and having your review included with the published papers. (If no, please inform the editor that you cannot review this manuscript.)YesIs the language of sufficient quality?YesPlease add additional comments on language quality to clarify if needed
Are all data available and do they match the descriptions in the paper? NoAdditional CommentsAccess to the raw data for the RNA-seq dataset that was used for gene predictions is not indicatedAre the data and metadata consistent with relevant minimum information or reporting standards? See GigaDB checklists for examples <a href="http://gigadb.org/site/guide" target="_blank">http://gigadb.org/site/guide</a>NoAdditional CommentsAny description of the RNA-seq dataset and its origin or features is fully missing.  I could not find other data that would be required according to guidelines in http://gigadb.org/site/guide: - Full (not summary) BUSCO results output files (text) - readme.txt including all file names with a brief description of each  - sample metadata that complies with the Genomic Standards ConsortiumIs the data acquisition clear, complete and methodologically sound?YesAdditional CommentsSequencing and bioinformatic methods followed are generally soundIs there sufficient detail in the methods and data-processing steps to allow reproduction?NoAdditional Comments1. Availability for the scripts used in bioinformatic analyses and data plotting is generally missing.   2. L124. Authors describe that minimap2 was used to obtain the dotplot. However, minimap2 alone does not produce dotplots.  3. L131. It is unclear how ‘PN40024’ 12X.v2, VCost.v3 protein annotations were used as input of BRAKER2. Do authors mean protein sequences instead? Where were these protein data retrieved from? How are proteins aligned to the assembly? Was BRAKER run from masked or unmasked asembly?Is there sufficient data validation and statistical analyses of data quality? NoAdditional Comments1. Validation of the original material for its true-to-typeness as 'Chambourcin' cultivar genotype is not mentioned, neither the number of different plants used for DNA extraction. While post-assembly validation of the Chambourcin genome assembly genotype from the mapped Chambourcin rhAmpSeq markers may be possible, such genotype validation is not mentioned either in the text.  2. In general, the quality and the genome variation represented in the Chambourcin genome assembly produced here could have been further tested. For instance, from 2% BUSCO duplication and 501.5 Mb of primary assembly size as compared to the 481.5 Mb haploid genome size that can be inferred from the k-mer analysis presented by the authors indicates, it seems that further duplication purging of the primary assembly is likely needed. This issue could be addressed by looking for assembly regions with reduced alignment depth when all HiFi reads are mapped to the primary assembly. Duplicated regions to be purged could also be supported by co-linear assembly segments sharing BUSCO duplicated genes. For assembly reliability assessment, 10X, rhAmpSeq, or Illumina WGS data that is available for Chambourcin could also be used to validate genome variants represented in this Chambourcin assembly when comparing the inter-haplotype variants detected between primary and haplotig assemblies or the haplotypes with genome assemblies from other genotypes 
Is the validation suitable for this type of data?YesAdditional CommentsThe validation is suitable, although it might not suffice in all cases.Is there sufficient information for others to reuse this dataset or integrate it with other data?NoAdditional CommentsAs described before, there is missing information at several instances, like for the origin of the RNA-seqAny Additional Overall Comments to the Author1. L171. Is it correct that total length of Bionano maps was as small as 962,964 bp? Or do authors mean kb instead of bp in that sentence?  2. The mapping of Chambourcin rhAmpSeq markers could have been further exploited to phase contig haplotypes before purging haplotypes and assembly scaffolding?  3. For the Conclusion in L254, it might be arguable whether the presented Chambourcin genome assembly is the first genome assembly of a complex interspecific hybrid or not. For instance 'Shine Muscat' might also be considered a complex inter-specific hybrid grape cultivar and its genome assembly was published: https://academic.oup.com/dnaresearch/article/29/6/dsac040/6808674 It might even be arguable whether the one presented in this publication is the first Chambourcin genome assembly as there is a 10X Genomics-based assembly available for Chambourcin: https://www.nature.com/articles/s41467-019-14280-1 
RecommendationMajor Revision